# SARS-CoV-2 replicates and displays oncolytic properties in clear cell and papillary renal cell carcinoma

Oi Kuan Choong[1,2], Rasmus Jakobsson[1,3], Anna Grenabo Bergdahl[4,5], Sofia Brunet[2,6], Ambjörn Kärmander[2,6], Jesper Waldenström[2,6], Yvonne Arvidsson[1,7], Gülay Altiparmak[1,7], Jonas A. Nilsson[2,8,9], Joakim Karlsson[2,8,9], Kristina Nyström[2,6]*, Martin E. Johansson[1,2,7]*

**1** Department of Laboratory Medicine, Sahlgrenska Center for Cancer Research, Institute of Biomedicine, University of Gothenburg, Gothenburg, Sweden, **2** Department of Laboratory Medicine, Institute of Biomedicine, The Sahlgrenska Academy, University of Gothenburg, Gothenburg, Sweden, **3** Department of Surgery and Urology, Skaraborg Hospital, Skövde, Sweden, **4** Department of Urology, Institute of Clinical Science, The Sahlgrenska Academy, University of Gothenburg, Gothenburg, Sweden, **5** Department of Urology, Sahlgrenska University Hospital, Gothenburg, Sweden, **6** Department of Clinical Microbiology, Sahlgrenska University Hospital, Gothenburg, Sweden, **7** Department of Clinical Pathology, Sahlgrenska University Hospital, Gothenburg, Sweden, **8** Department of Surgery, Sahlgrenska Center for Cancer Research, Institute of Clinical Sciences, The Sahlgrenska Academy, University of Gothenburg, Gothenburg, Sweden, **9** Harry Perkins Institute of Medical Research, University of Western Australia, Perth, Australia

* martin.e.johansson@gu.se (MEJ); kristina.nystrom@microbio.gu.se (KN)

**Data Availability Statement:** All relevant data are within the manuscript and its Supporting information files.

## Abstract

The SARS-CoV-2 virus is currently causing a global pandemic. Infection may result in a systemic disease called COVID-19, affecting primarily the respiratory tract. Often the gastrointestinal tract and kidneys also become involved. Angiotensin converting enzyme 2 (ACE2) serves as the receptor for SARS-CoV-2. The membrane proteins, Transmembrane serine protease 2 (TMPRSS2) and Neuropilin 1 (NRP1) are accessory proteins facilitating the virus entry. In this study we show that the human proximal kidney tubules, express these factors. We hypothesized that cancers derived from proximal tubules as clear cell (CCRCC) and papillary renal cell carcinoma (PRCC), retain the expression of the SARS-CoV-2 entry factors making these cancers susceptible to SARS-CoV-2 infection. We used bioinformatics, western blotting, and assessment of tissue micro arrays (TMA) including 263 cases of CCRCC, 139 cases of PRCC and 18 cases of chromophobe RCC to demonstrate that the majority of CCRCC and PRCC cases retained the RNA and protein expression of the entry factors for SARS-CoV-2. We furthermore show that SARS-CoV-2 virus propagated robustly in primary cultures of CCRCC and PRCC cells with a visible virus cytopathogenic effect correlating with viral RNA expression levels. We also noted that the delta-variant of SARS-CoV-2 causes cancer cells to form syncytia *in-vitro*. This phenomenon was also identified histologically in CCRCC tissue from a patient that had been hospitalized for COVID-19, twelve months prior to nephrectomy. Our data provide insights into SARS-CoV-2 infectivity in renal cell carcinoma and that the virus causes a distinct cytopathogenic effect.

**Funding:** MJ: Grant number 21 1767 Pj 01 H from The Swedish Cancer Society (Cancerfonden) MJ: The National Association against Kidney Diseases. MJ: Grant number 71390 Swedish Government Funding of Clinical Research within the National Health Service (ALF), MJ: The strategic research programme BioCARE, RJ: Grant number 273289 The Research Fund (R&D) at Skaraborg Hospital, Skövde, Sweden and the Healthcare Committee, Region Västra Götaland, Sweden. The funders had no role in study design, data collection and analysis, decision to publish, or preparation of the manuscript.

**Competing interests:** The authors have declared that no competing interests exist.

# Introduction

In 2019 a novel coronavirus was identified in China. It was subsequently named Severe Acute Respiratory Syndrome Coronavirus 2 (SARS-CoV-2) and has the capacity to cause a severe respiratory and systemic disease, called Corona Virus Disease 2019 (COVID-19). SARS-CoV-2 is a positive-sense single-stranded RNA virus. It has been shown that the virus uses the plasma membrane protein Angiotensin Converting Enzyme 2 (ACE2) as receptor to enter the host cell [1, 2]. Virus uptake has also been shown to depend on proteolytic cleavage and priming of the viral spike proteins by enzymes such as FURIN, Transmembrane protease serine 2 (TMPRSS2) or Cathepsin L (CATL). Recently it has also been reported that Neuropilin 1 (NRP1) constitutes an important final viral entry factor by enhancement of the ACE2 binding by the primed spike protein [3]. Co-expression of *ACE2* and *TMPRSS2* mRNA has been reported in cornea, nasopharynx, lung and gastrointestinal tissues [2]. COVID-19 is indeed a systemic disease however and SARS-CoV-2 has also been shown to cause renal impairment, with several potential mechanisms of injury, including direct kidney tropism [4]. It has previously been shown that FURIN is expressed by the tubular cells of the human kidney [5] and *ACE2*, *NRP1* and *TMPRSS2* mRNA and protein have been shown to be expressed by human kidney tissue [6–8]. A previous bioinformatical study has shown that esophageal squamous cell carcinoma, cervical cancer and papillary renal cell carcinoma express *ACE2* and *TMPRSS2* mRNA [9]. The two major forms of renal cell carcinoma (RCC), clear cell renal cell carcinoma (CCRCC) and papillary renal cell carcinoma (PRCC) represent about 90% of all RCC. Both originate from the proximal tubules, whereas the third most common RCC, chromophobe RCC (CHRCC) stems from the cortical collecting ducts [10]. Recently introduced targeted therapies offer ways to slow the disease progression, but surgery remains the only curative treatment. In this study we investigated the hypothesis that the expression of the key viral entry factors may be retained by CCRCC and PRCC thereby rendering the RCC cells susceptible to SARS-CoV-2 infection.

# Materials and methods

## Tissue procurement and primary cell culture

Tissue for primary cell culture and protein analysis was obtained from nephrectomies performed due to kidney malignancy and in compliance with ethical regulations and following informed patient consent. The patients were treatment naïve and had not received any treatment prior to surgery performed on curative intent. Ethical approval was granted by the Swedish Ethical Review Authority (LU680-08, LU289-07, 2019–00905). Clinical pathological data of the included cases are shown in the S1 Table.

For normal renal tubular cell culture, macroscopically normal tissue was sampled farthest from the tumor site. Cancer tissue was chosen from areas where the tumor seemed clearly viable and without signs of necrosis. Both normal kidney and tumor tissue was confirmed histologically by an experienced urological pathologist. Tissues were collected in serum-free DMEM medium (GE Healthcare, Little Chalfont, UK) supplemented with 1% penicillin-streptomycin (GE Healthcare) and transferred on ice to the cell culture facility. Tissue samples were minced into smaller pieces with scissors and digested overnight in full DMEM medium containing Collagenase Type I (300 U/ml, Thermo Fischer Scientific, Waltham, MA, USA) and DNase I, type II (200U/ml, Sigma Aldrich, St Louis, MO, USA). Following incubation, the digested tissues were treated with 0.125% Trypsin (GE Healthcare) and triturated through a sterile 5 ml pipette. Thereafter cells were sequentially passed through 40 and 20 μm strainers to ensure a single cell solution. Cells were cultured in DMEM supplemented with 1% penicillin-

streptomycin and 10% fetal calf serum, at 37˚C, 5% $CO_2$ in a humidified incubator. Samples for preparation of lysates for Western blotting were chosen from the same tissue areas as those for primary culture.

## Bioinformatics and gene expression analysis

RNA-seq read counts for 9724 primary tumors from 32 TCGA cohorts, calculated with HTSeq based on alignments to the hg38 human reference genome assembly, were downloaded using TCGAbiolinks (v. 2.17.1) [11] in R (v. 3.6.1) and normalized to RPKM values relative to the longest reference transcript of each gene using the *rpkm* function in edgeR (v. 3.28.1) [12] with the parameters *log = F*, *prior.count = 0*. Transcript lengths were obtained using *biomaRt* (v. 2.42.1) [13]. In box plots, edges of boxes represent the first and third quartiles, whiskers represent the smallest/largest data points at most 1.5 times inter-quartile range from the lower and upper bound, respectively.

## Tissue micro arrays and histological analysis

Tissue micro arrays (TMA) of clear cell, papillary and chromophobe renal cell carcinoma were used to assess the histological expression of ACE2, TMPRSS2 and NRP1. The CCRCC TMA was constructed from 263 cases of CCRCC. From 30 of these cases, material from metachronous metastases was also included. The papillary renal cell carcinoma TMA was assembled from 139 cases of PRCC and the TMA from chromophobe RCC was constructed from 18 cases of chromophobe renal cell carcinoma. For all three TMAs, cases were re-evaluated by an experienced urological pathologist (M.J) and representative areas were marked for inclusion. Two 1 mm punches per case were sampled and added to the recipient block. The TMA blocks were sectioned at 3 μm. Histological material for Hematoxylin/Eosin staining and immunohistochemistry was sectioned at 3 μm and stained according to standard procedure. For immunohistochemistry, sections were pretreated using a Dako PT-Link with EnVision FLEX Target Retrieval Solution (high pH). Immunohistochemistry was performed using a Dako Autostainer Link with EnVision FLEX reagents according to the manufacturer's instructions (DakoCytomation, Glostrup, Denmark). The RCC4 cell line was paraffin embedded using the Cellient automated cell block system according to the manufacturer´s instructions (Hologic, Mississauga, Canada). Antibodies used for immunohistochemistry were rabbit anti-ACE2 (1:500, HPA000288, Atlas antibodies, Stockholm, Sweden), mouse anti-TMPRSS2 (1:500, sc-515727, Santa Cruz Biotechnology, Texas, United States) and rabbit anti-NRP1 (1:4000, ab81321, Abcam, Cambridge, United Kingdom).

## Western blotting

Approximately 50mg of tissue samples were lysed in RIPA lysis buffer (Millipore, Massachusetts, United States) supplemented with protease inhibitor (Sigma Aldrich, St Louis, MO, United States) by TissueLyser II (Qiagen, Hilden, Germany). Total protein was collected and quantified using Pierce BCA protein assay kit (23225, Thermo Fisher Scientific, MA, United States). Protein samples were separated in NuPAGE Bis-Tris gel electrophoresis system (ThermoFisher Scientific, Massachusetts, United States) and transferred to PVDF membrane. The membranes were incubated with mouse anti-TMPRSS2 (1:1000, MA5-35756, Thermo Fisher Scientific, MA, United States), rabbit anti-ACE2 (1:1000, HPA000288, Atlas antibodies, Stockholm, Sweden), rabbit anti-NRP1 (1:1000, ab81321, abcam, Cambridge, United Kingdom), mouse anti-β actin (1:5000, ab8226, abcam, Cambridge, United Kingdom), respectively. The membranes were detected and imaged using Amersham ImageQuant 800 system (Cytiva, Uppsala, Sweden).

## Primary cell exposure to SARS-CoV-2 virus

The SARS-CoV-2 DE-Gbg20 strain was used (MW092768). Virus was cultured on Vero cells (ATCC CCL-81). The Vero cells were cultured in DMEM supplemented with 2% heat-inactivated FCS, and 100 U of penicillin and 60 μg/ml of streptomycin during virus infection. The virus was titrated on Vero cells and 100 TCID50 was used for infection. Virus was added to each well of 24 well plates of three individual normal kidney, three CCRCC and three PRCC cell cultures in triplicate and the infection was monitored for cytopathic effect (CPE). 100ul supernatant was collected from wells at 0, 24, 48 and 72 hours.

The DE-Gbg20 strain and the delta variant, as confirmed by NGS sequencing, cultured in Vero cells, were used to infect normal kidney cells and CCRCC from three individuals. 100 and 1000 TCID50 were used for infection in triplicate as well as mock-infected cells. CPE was monitored every 24 hours and photographed using a CytoSMART Lux2 microscope (CytoS-MART Technologies, Eindhoven, The Netherlands).

## RNA extraction and real time PCR

Supernatant was lysed in RLT lysis buffer (Qiagen, Hilden, Germany) according to the manufacturer's instructions and the samples were further heat-inactivated at 56˚C for one hour. RNA was isolated using RNeasy mini protocol (Qiagen, Hilden, Germany). The real-time PCR was performed on a 7300 Real-Time PCR machine (Applied Biosystems) and all samples were tested in triplicate. The reaction was performed in a 20 μl reaction mixture containing Superscript III Platinum One-Step qRT-PCR Kit (Invitrogen) and 0.3 μM of each primer, forward primer GTCATGTGTGGCGGTTCACT and reverse primer CAACACTATTAGCATAAG-CAGTTGT and 0.2 μM of probe [FAM] CAGGTGGAACCTCATCAGGAGATGC [BHQ1], all located in the RdRp of SARS-CoV-2. The qPCR was initiated with reverse transcription at 46˚C for 30 min followed by one cycle of 95˚C for 10 min and 45 cycles of 95˚C for 15 sec and 56˚C 1 min. A plasmid containing the target sequence was used as a control in four 10-fold serial dilutions and from this the viral genome copies per ml were calculated.

## Histological analysis of CCRCC from a patient previously hospitalized for COVID-19

The Department of Urology at Sahlgrenska University Hospital performed a search for patients that had undergone nephrectomy due to renal cell carcinoma with a previously confirmed SARS-CoV-2 infection. One case was identified. The patient was male, 52 years of age and had been hospitalized for COVID-19, 10 months prior to surgery. Infection had been confirmed by PCR and subsequent positive serology. Analysis of the histological material was performed by an experienced urological pathologist (M.J).

# Results

## Expression of ACE2, TMPRSS2 and NRP1 is distinct in clear cell and papillary renal cell carcinoma and ACE2 and NRP1 display the highest mRNA expression across 32 distinct cancer forms

To test the hypothesis that the SARS-CoV-2 virus displays tropism not only to kidney cells, but also to renal cell carcinoma, we utilized the publicly available transcriptomic data sets deposited by The Cancer Genome Atlas project (TCGA) to assess the transcriptional levels of *ACE2*, *TMPRSS2* and *NRP1* mRNA in individual CCRCC, PRCC and CHRCC cases. The results were grouped according to RCC type (Fig 1A). The mRNA levels for *ACE2* and *NRP1* proved to be high in both CCRCC and PRCC, whereas levels for *TMPRSS2* were lower, but significant.

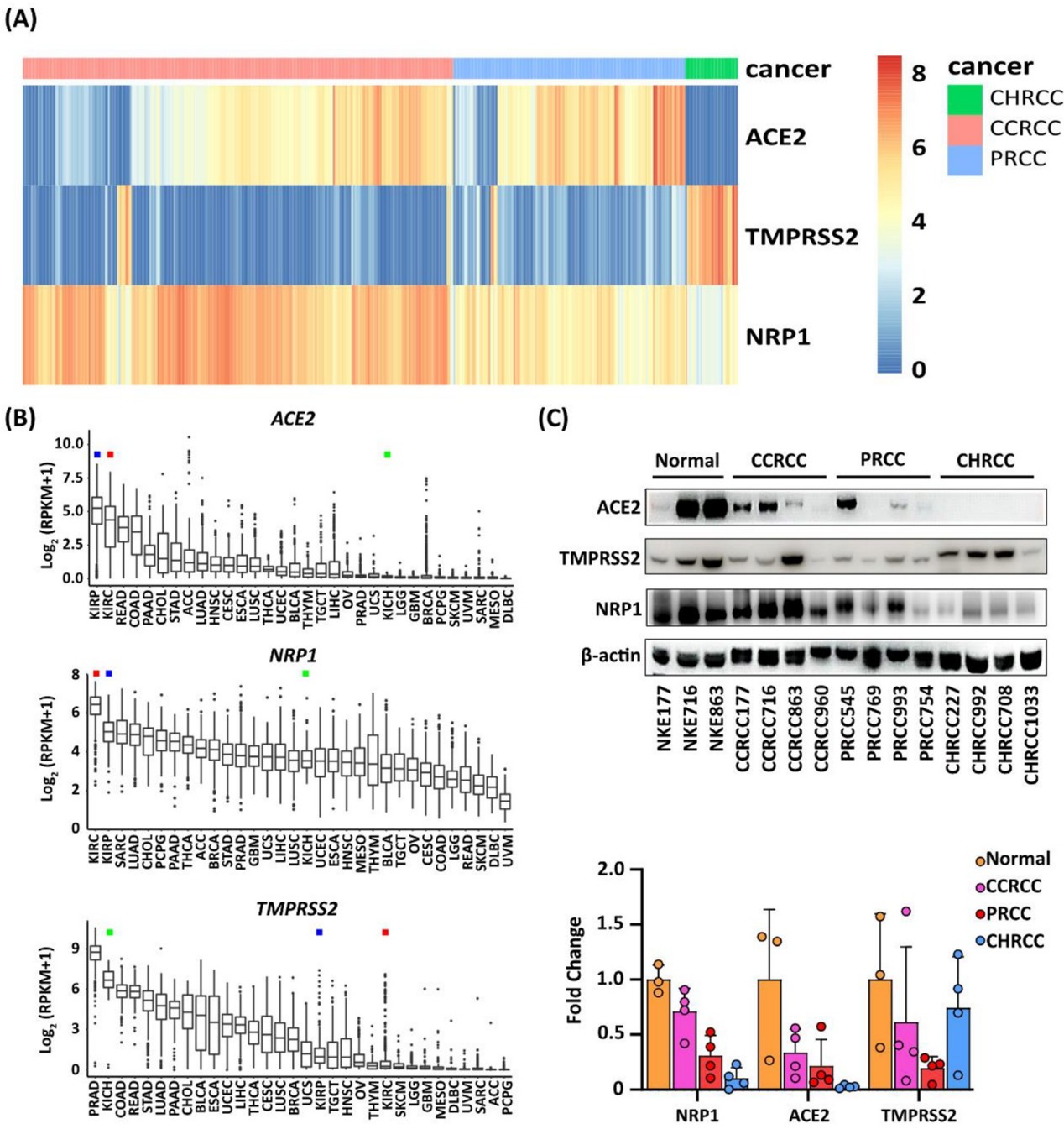

**Fig 1. Expression of ACE2, TMPRSS2 and NRP1 in clear cell, papillary and chromophobe renal cell carcinoma and across 32 distinct cancer forms.** (A) Bioinformatic analysis of the TCGA data sets for renal cell carcinoma showing mRNA expression levels for the *ACE2*, *TMPRSS2* and *NRP1* genes in individual RCC cases for chromophobe renal cell carcinoma (CHRCC), clear cell renal cell carcinoma (CCRCC) and papillary renal cell carcinoma (PRCC). (B) Comparison of the mRNA expression levels for *ACE2*, *TMPRSS2* and *NRP1* across 32 different types of malignancies deposited at the TCGA atlas. KIRC denotes CCRCC (red square), KIRP denotes PRCC (blue square) and KICH denotes CHRCC (green square). (C) Immunoblotting results for ACE2, TMPRSS2 and NRP1 proteins in lysates from normal tissue, CCRCC, PRCC and CHRCC. Results from densitometry of the protein bands is shown below the blots.

The reverse relation was seen for CHRCC where *ACE2* and *NRP1* levels were low but *TMPRSS2* highly expressed. We furthermore analyzed the transcriptional levels of *ACE2*, *TMPRSS2* and *NRP1* mRNA across 32 different types of malignancies deposited at the TCGA. Of all cancer forms analyzed, CCRCC and PRCC displayed the highest expression levels for *ACE2* and *NRP1* mRNA, whereas again *TMPRSS2* expression was lower but not negligible (Fig 1B). To elucidate the expression of the actual proteins, we performed western blot analysis of protein lysates from normal kidney cortex (*n* = 3), CCRCC tissue (*n* = 4), PRCC tissue (*n* = 4) and CHRCC tissue (*n* = 4). We discovered that ACE2, TMPRSS2 and NRP1 protein was detected in normal cortical tissue, CCRCC and PRCC (Fig 1C). The ACE2 protein was clearly detectable by three out of four CCRCC and PRCC cases and was undetectable in CHRCC lysates. NRP1 expression was strongly expressed by three of four CCRCC and PRCC cases, and at lower intensity by chromophobe RCC. Samples from CCRCC, PRCC and CHRCC expressed TMPRSS2 protein.

## ACE2, TMPRSS2 and NRP1 expression by proximal kidney tubules is retained by clear cell and papillary renal cell carcinoma tissue

To allow for high-throughput analysis of ACE2, TMPRSS2 and NRP1 protein expression in RCC, we stained tissue micro arrays (TMA) with 263 cases of CCRCC, 139 cases of PRCC, 18 cases of CHRCC and human kidney tissue as control with antibodies against ACE2, TMPRSS2 and NRP1. We found that all three proteins were localized to the apical plasma membranes of the proximal tubules of normal human kidney (Fig 2A). ACE2 and NRP1 could not be detected in other tubular segments, whereas a faint TMPRSS2 signal was detected in the collecting ducts. Clear cell renal cell carcinoma and papillary renal cell carcinoma both originate from the proximal tubules, whereas chromophobe renal cell carcinoma originates from the intercalated cortical collecting duct cells. Of the CCRCC cases, 76%, 81% and 85% were positive for ACE2, TMPRSS2 and NRP1 respectively. PRCC displayed 93%, 56% and 66% positivity for ACE2, TMPRSS2 and NRP1 (Fig 2B and 2C, Table 1). CHRCC was negative for ACE2 and NRP1 but displayed a faint TMPRSS2 positivity in 50% of the cases, (Fig 2D, Table 1). Cells from the CCRCC derived cell line RCC4 were also fixed by formalin and paraffin embedded. No ACE2, TMPRSS2 and NRP1 staining was detected in the plasma membranes of the RCC4 cells. A distinct nuclear signal for ACE2 was however noted, (S1 Fig).

## SARS-CoV-2 virus replicates in cultured renal cell carcinoma cells and causes a distinct virus cytopathogenic effect

Having established that the key receptor and cofactors for SARS-CoV-2 binding were expressed by normal kidney, CCRCC tissue and PRCC tissue, we proceeded to investigate if primary RCC cancer cells, primary normal tubular cells and the CCRCC cell line RCC4 would internalize and propagate SARS-CoV-2 virus. The RCC4 cell line was included in order to evaluate if cell lines retain entry factor expression. We used primary RCC cells from the same cases that had been previously assessed by western blotting for expression of the entry factors. Primary cultures from 3 individual normal kidneys, 3 CCRCC cases and 3 PRCC cases were established. The cells were subsequently exposed to SARS-CoV-2 virus and propagation was monitored by virus negative strand PCR and visual inspection of the cultures at 24, 48 and 72 hours. Results showed that SARS-CoV-2 infected and propagated in primary normal kidney cells, CCRCC cells and PRCC cells as determined by real-time PCR and visual inspection. The RCC4 cell line failed to demonstrate infection detectable by PCR (Fig 3A). One normal kidney culture (NKE863), one CCRCC (CCRCC863) and two PRCC cell cultures (PRCC545 & PRCC769) also displayed distinct visible signs of virus cytopathogenic effect (CPE) at 48 hours

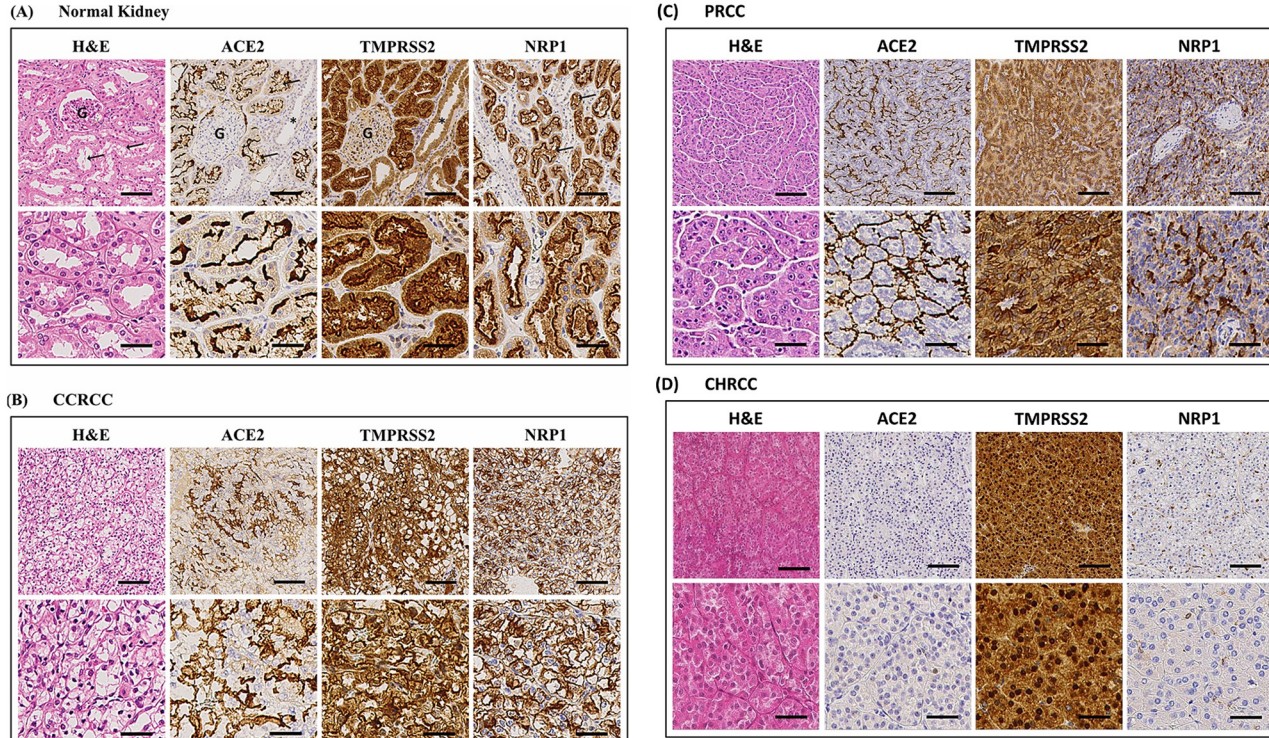

**Fig 2. Expression of ACE2, TMPRSS2 and NRP1 in normal kidney and renal cell carcinoma tissue.** Haematoxylin & eosin staining of tissues and immunohistochemistry staining for ACE2, TMPRSS2 and NRP1. (A) Normal kidney displays staining of the apical plasma membranes of the proximal tubules for all three proteins. G = glomerulus, arrows = examples of proximal tubules and asterisk = collecting duct (negative). (B) Clear cell renal cell carcinoma tissue shows a diffuse staining of the plasma membranes for ACE2, TMPRSS2 and NRP1. (C) Papillary renal cell carcinoma displays an apical staining pattern for all three proteins. The lower rows of the panels show higher magnification of the images above. (D) Chromophobe renal cell carcinoma displayed negative staining for ACE2 and NRP1 whereas TMPRSS2 was strongly positive in the cancer cells. Upper panel scale bar = 100μm, lower panel scale bar = 40μm. (CCRCC = clear cell renal cell carcinoma, PRCC = papillary renal cell carcinoma, CHRCC = chromophobe renal cell carcinoma).

(Fig 3B) and 72 hours post infection. This also correlated with the highest expression of viral RNA (Fig 3A). Additionally, one of the SARS-CoV-2 infected CCRCC cell cultures (CCRCC716) resulted in a moderate CPE. This culture also displayed slightly lower RNA levels, than in cultures where a clear CPE was visible (Fig 3A). Two normal kidney cultures (NKE716 & NKE177), one CCRCC (CCRCC960) and one PRCC (PRCC993) showed a limited increase in viral RNA and no CPE was visible. However, one of these normal kidney cultures demonstrated high expression of ACE (Fig 1C).

**Table 1. Assessment of protein expression of ACE2, TMPRSS2 and NRP1 in clear cell, papillary and chromophobe renal cell carcinoma.**

| PROTEIN | CCRCC (N = 263) | TMA PRCC (n = 139) | CHRCC (N = 18) |
|---|---|---|---|
| ACE2 | **76%** (201/263) | **93%** (125/134) | **0%** (0/18) |
| TMPRSS2 | **81%** (207/255) | **56%** (74/131) | **50%** (9/18) |
| NRP1 | **85%** (220/258) | **66%** (91/137) | **0%** (0/18) |

TMA, tissue microarray; CCRCC, clear cell renal cell carcinoma; PRCC, papillary renal cell carcinoma; CHRCC, chromophobe renal cell carcinoma. Due to random drop out of TMA-cores the number of assessed cores may vary slightly.

**(A)**

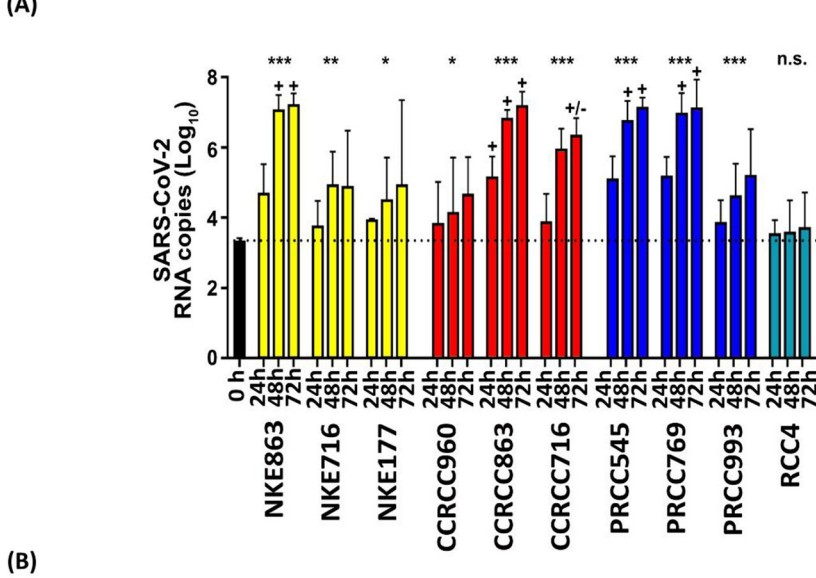

**(B)**

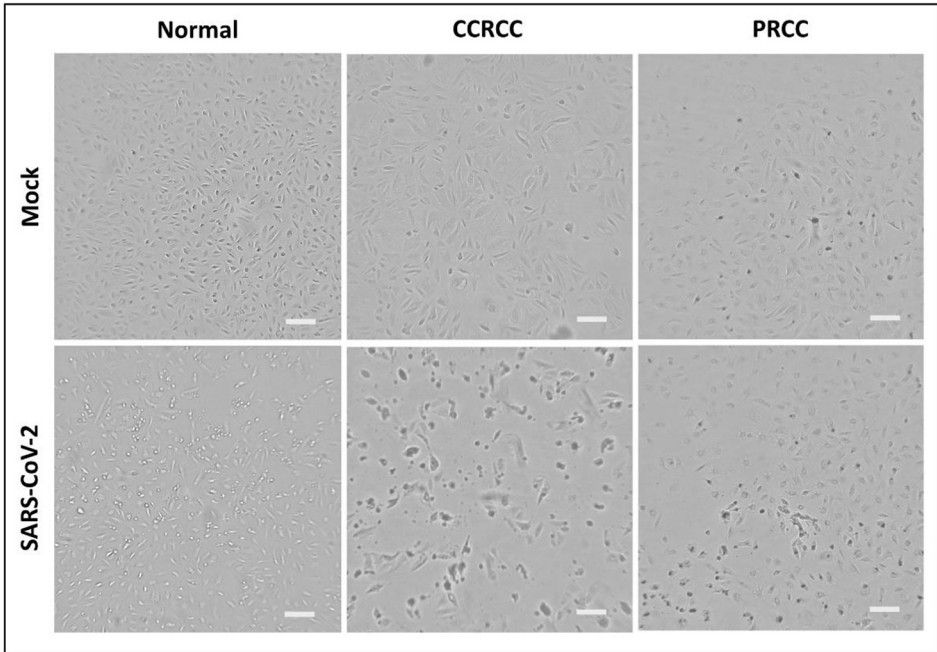

**Fig 3. SARS-CoV-2 virus replicates in cultured renal cell carcinoma cells and causes a distinct virus cytopathogenic effect.** (A) Negative strand qPCR analysis of SARS-CoV-2 viral RNA copy number at 24h, 48h and 72h post-infection in primary normal tubular epithelial cells, CCRCC cells, PRCC cells and the renal cell carcinoma cell line RCC4. (B) Representative images of virus cytopathogenic effect (CPE) for normal (NKE863), clear cell renal cell carcinoma (CCRCC) (CCRCC863) and papillary renal cell carcinoma (PRCC) (PRCC545) at 48 hours after infection. The CPE causes rounding up of cells, followed by detachment from the growth surface. Mock is control. Scale bar = 200μm.

## Unusual histological features in clear cell renal cell carcinoma tissue from a patient with previous COVID-19

Patients with COVID-19 are not candidates for nephrectomy. We however identified one CCRCC case from our records where the patient had undergone a serologically confirmed and

hospital treated COVID-19 episode 12 months prior to nephrectomy. Histologically, the case was a clear cell carcinoma with very unusual features (S2A Fig). About 80% of the tumor was necrotic (S2B Fig). The remaining part displayed odd features of cancer cell fusion where the CCRCC cells had fused into syncytia (S2C and S2D Fig). The cancer cells in these areas also were discohesive and rounded with signs of degeneration. PCR for SARS-CoV-2 proved negative in the cancer tissue.

## Visible formation of syncytia in primary CCRCC cells exposed to the delta virus variant

Following up on the finding of potential syncytial changes in SARS-CoV-2 exposed CCRCC tissue we investigated the *in vitro* effects of two SARS-CoV-2 variants, D614G & Delta. These were inoculated with normal renal tubular cell and CCRCC primary cell cultures, respectively (S3 Fig). Formation of CCRCC cell syncytia was clearly observable after 48 hours of incubation in clear cell renal cell carcinoma cells infected by the Delta variant *in vitro* (S3B Fig).

## Discussion

COVID-19 is the systemic syndrome caused by severe infection by the SARS-CoV-2 virus. The virus enters the host via the respiratory tract, where many cell types express the receptors and proteins necessary for virus uptake. The most important of these proteins is the actual receptor for the virus, the ACE2 protein. Cofactors such as NRP1 and TMPRSS2 have been shown to be crucial as well however. ACE2 is relatively broadly expressed and the gastrointestinal tract expresses *ACE2* mRNA in orders of magnitude above the respiratory tract levels [14]. Co-expression of receptor and co-factors is seen in a relatively small number of tissues and organs however. The gastrointestinal tract has been shown to be a site for active viral uptake. Furthermore, kidney injury is often diagnosed during COVID-19 disease. This was initially regarded as secondary to major systemic illness, but since then reports of direct renal infection have been published. We show that the necessary proteins for viral uptake, indeed are expressed by the proximal tubules of the human kidney. The proximal tubular cells are cells of origin for CCRCC and PRCC and thus for the major types of RCC. Importantly, we found that the expression is retained in a majority of CCRCC and PRCC cases, as assessed by immunohistochemistry combined with bioinformatical analysis of tumor mRNA levels for these genes deposited in the TCGA data sets. We found that CCRCC and PRCC clearly stand out in the TCGA data sets. The expressional levels for *ACE2* and *NRP1* mRNA occupy first and second position across all cancers analysed for CCRCC and PRCC, whereas *TMPRSS2* was more modestly expressed. We could furthermore demonstrate that the virus replicated and displayed a clear virus cytopathogenic effect in a majority of the primary cancer cells. Uptake and effect were not always clearly correlated with ACE2 expression levels as determined by western blot. This may indicate additional factors playing a role in causing cytopathogenic effects in cell culture. The effect on the primary RCC cells was similar in magnitude to that of the kidney derived cell lines used to propagate SARS-CoV-2 virus in the clinical setting at the department of virology. Other cell lines used, such as calu3 require higher viral load for CPE and high replication. Virus replication was not shown to occur in the RCC4 cell line that proved to be negative for plasma membrane bound ACE2 and cofactors. This could be an argument that primary RCC cells are required to study this phenomenon. We are aware of the fact that the present paper describes *in-vitro* data, albeit based on primary cancer cells. We sought to amend this by identifying all patients with confirmed COVID-19 that had undergone a subsequent nephrectomy due to malignancy. It turned out that this patient category is quite small, since only one case was identified, where the COVID-19 diagnosis was made on clinical

grounds and confirmed by serology. Histological assessment showed that the cancer tissue displayed very unusual features. The viable cancer cells displayed a rounded-up morphology and discohesive growth pattern with extensive necrotic zones. In areas cancer cells had fused into peculiar multinuclear syncytial cells, again with discohesive growth pattern. The RCC tissue was negative regarding SARS-CoV-2 RNA, but this is as expected since the COVID-19 episode predated the surgery with 12 months. It however led us to investigate *in vitro* effects of SARS-CoV-2 variants on CCRCC cells. We could observe a distinct syncytialization effect on the CCRCC cells following exposure to the delta variant. We are aware of the anecdotal nature of this RCC case and can therefore not draw any conclusions regarding potential effects in human RCC. We suggest however that this patient category may be important to follow in order to establish a potential causal connection where virus infection of the cancer cells may cause tumor lysis in the patient. In agreement with this reasoning, a recent study has shown that three patients with colorectal cancer did observe tumor regression while suffering from Covid-19 [15]. Recently it has been found that not only the actual oncolytic cellular effect is of value. Of similar or greater importance is the possibility of an enhanced immune response elicited by the cancer cells undergoing viral infection [16]. The virally affected cells release antigens that can be recognized by the immune system. This approach has become interesting as an adjunct to immune-oncological treatment protocols. At present, there is only one registered viral oncolytic treatment: talimogene laherparepvec (Imlygic). It is based on an attenuated herpes virus that has been modified to selectively replicate in melanoma cells and to stimulate a T-cell mediated immune response. It is administered by direct injection into the tumor [17]. Whether SARS-CoV-2 uptake and propagation in RCC cells could elicit a similar response is uncertain. The isolated case described above does not prove causality for SARS-CoV-2 induced effects on CCRCC in patients. Furthermore, usage of this uptake mechanism is questionable due to the pathogenicitity of the virus and lack of selectivity due to expression in normal organs. We conclude however that SARS-CoV-2 infects and displays virus cytopathogenic effects in clear cell and papillary renal cell carcinoma *in-vitro*. The cytopathogenic effect causes RCC cell lysis and the delta-variant of the virus causes syncytialization of the cancer cells.

## Supporting information

**S1 Fig. Expression of ACE2, TMPRSS2 and NRP1 in the RCC4 cell line.** None of the proteins could be detected at the expected location in the CCRCC cell line RCC4. ACE2 displayed nuclear positivity, NRP1 a faint cytoplasmic staining, whereas TMPRSS2 was negative. Upper panel scale bar = 100μm, lower panel scale bar = 40μm.
(TIF)

**S2 Fig. Unusual histology in clear cell renal cell carcinoma from a COVID-19 patient.** (A) The basic histopathological pattern of the cancer case was clear cell renal cell carcinoma. (B) In about 60% of the tumor tissue extensive areas of necrosis could be seen. (C and D) Areas displaying very unusual features where the cancer cells had coalesced into syncytial, multinuclear and discohesive cancer cells with fusion of the lipid laded cytoplasms. All images were stained by Hematoxylin/Eosin, Scale bars in A-C: 100 μm, in D: 200 μm.
(TIF)

**S3 Fig. *In-vitro* formation of syncytia in primary CCRCC cells exposed to virus variants.** Representative images of (A) normal tubular epithelial cells and (B) clear cell renal cell carcinoma cells after exposure to the SARS CoV-2 variants D614G or delta. Uninfected is mock control. After 48 hours of exposure to the delta variant a distinct formation of syncytia is seen

in the CCRCC cultures. Hatched lines mark syncytial cancer cells. Scale bar = 200μm.
(TIF)

**S1 Raw images. Uncropped western blot.**
(ZIP)

**S1 Table. Clinical characteristics of the patients included in the study.**
(DOCX)

## Author Contributions

**Conceptualization:** Martin E. Johansson.

**Data curation:** Martin E. Johansson.

**Formal analysis:** Oi Kuan Choong, Rasmus Jakobsson, Kristina Nyström.

**Investigation:** Oi Kuan Choong, Rasmus Jakobsson, Kristina Nyström.

**Methodology:** Oi Kuan Choong, Rasmus Jakobsson, Anna Grenabo Bergdahl, Sofia Brunet, Ambjörn Kärmander, Jesper Waldenström, Yvonne Arvidsson, Gülay Altiparmak, Joakim Karlsson, Kristina Nyström.

**Resources:** Anna Grenabo Bergdahl, Sofia Brunet, Ambjörn Kärmander, Jesper Waldenström, Jonas A. Nilsson, Martin E. Johansson.

**Software:** Oi Kuan Choong, Joakim Karlsson, Kristina Nyström.

**Supervision:** Martin E. Johansson.

**Validation:** Oi Kuan Choong, Rasmus Jakobsson, Yvonne Arvidsson, Gülay Altiparmak, Kristina Nyström.

**Visualization:** Oi Kuan Choong, Martin E. Johansson.

**Writing – original draft:** Martin E. Johansson.

**Writing – review & editing:** Oi Kuan Choong, Rasmus Jakobsson, Anna Grenabo Bergdahl, Sofia Brunet, Ambjörn Kärmander, Jesper Waldenström, Yvonne Arvidsson, Jonas A. Nilsson, Joakim Karlsson, Kristina Nyström, Martin E. Johansson.

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
