## [Decision Letter · Decision Letter 0]

4 Jul 2022

PONE-D-22-06970SARS-CoV-2 replicates and displays oncolytic properties in clear cell and papillary renal cell carcinomaPLOS ONE

Dear Dr. Johansson,

Thank you for submitting your manuscript to PLOS ONE. After careful consideration, we feel that it has merit but does not fully meet PLOS ONE’s publication criteria as it currently stands. Therefore, we invite you to submit a revised version of the manuscript that addresses all comments raised by the two reviewers and in particular their comments on syncytia formation and the mechanisms underlying the described cytopathic effects.

We look forward to receiving your revised manuscript.

Kind regards,

Birke Bartosch

Academic Editor

PLOS ONE

https://journals.plos.org/plosone/s/file?id=ba62/PLOSOne_formatting_sample_title_authors_affiliations.pdf".

“This study was supported by grants from the Swedish Cancer Society (Cancerfonden), the National Association against Kidney Diseases, Swedish Government Funding of Clinical Research within the National Health Service (ALF), the strategic research program BioCARE, the Research Fund (R&D) at Skaraborg Hospital, Skövde, Sweden and the Healthcare Committee, Region Västra Götaland, Sweden.”

“RJ: the Healthcare Committee, Region Västra Götaland, Sweden - 273289.

Reviewers' comments:

Reviewer's Responses to Questions

**Comments to the Author**

1. Is the manuscript technically sound, and do the data support the conclusions?

Reviewer #1: No

Reviewer #2: Yes

2. Has the statistical analysis been performed appropriately and rigorously? 

Reviewer #1: N/A

Reviewer #2: Yes

3. Have the authors made all data underlying the findings in their manuscript fully available?

Reviewer #1: Yes

Reviewer #2: Yes

4. Is the manuscript presented in an intelligible fashion and written in standard English?

Reviewer #1: Yes

Reviewer #2: Yes

5. Review Comments to the Author

Reviewer #1: The study presented by Choong et al. implies that SARS-CoV-2 may replicate in renal cell carcinoma cells (CCRCC and PRCC) because these they express virus entry factors. Infection with SARS-Cov-2 resulted in cytopathic effect – syncytia formation. The manuscript is well written but I am afraid that some of the data presented suffer flaws and at some points there is overinterpretation of the results.

Here are my comments concerning the study:

1. Line 71: the authors conclude that their data “provide insights into SARS-CoV-2 infectivity in renal cell carcinoma and define a potential viral entry mechanism that could be used to target renal carcinoma”. However, the latter is not shown experimentally nor sufficiently discussion in the Discussion section. Moreover, in the Discussion section, the authors widely stress on oncolytic viral therapy implicating SARS-CoV-2. Such notion is not supported experimentally. Do cancer cells strongly overexpress viral entry factors compared to normal cells? How will be the specific targeting of cancer cells achieved? Indeed, measles virus (the vaccine strain!) has been used in such therapies, but this is justified by the strong expression of MCP (CD46) in cancer cells and the non-pathogenicity of the virus which is not the case with SARS-CoV-2.

2. Concerning the Discussion, the authors are not the first to show that SARS-CoV-2 entry factors are found in renal cells. This are just some examples of other studies: PMID:32903321 (TMPRSS2), PMID:19736548, PMID:11566082 (NRP1), and for ACE-2 there are multiple papers.

3. Figure 1C: I am afraid I do not see the expression of TMPRSS2 in normal tissues on the blot as stated by the authors: line 226. The authors also stated that “NRP1 expression was robust across all cases” (Line 228). This is not the case for CHRCC and some of PRCC samples. A better blot must be provided, and also a quantification of the blot by densitometry.

4. Figure 2A: Use arrows to indicate different histological structures.

5. Line 246: “We also formalin” The verb is missing…

6. Figure 3B: The data of syncytia formation in PRCC is not convincing (the microscopic image). Moreover, there is no correlation between the cell culture data (slightly visible syncytia) and the PCR data (strong expression) for PRCC.

7. I do not understand the including of the case study of 1 patient within this report. There is absolutely no proof that syncytia formation is due to SARS-CoV-2 (that the patient had 1 year before). Research for viral antigens was not performed. Moreover, syncytia formation is directly linked to cell lysis. This is even seen in the in vitro data on FigS3 B (almost no cells left).

8. In the discussion section (Lines 312-316) the authors say that “uptake and effect were not always clearly correlated to ACE-2 expression and that additional factors play role in causing cytopathogenic effect (CPE) in cell culture”. CPE is usually due to expression of viral receptors on uninfected cells and the expression of viral glycoproteins in infected cells. What is the expression of spike in these cells?

Reviewer #2: The manuscript by Choong et al describes a very promising phenomenon: oncolytic activity of SARS-CoV-2 against renal carcinoma variants. The authors used bioinformatic analysis of existing RNAseq data and showed that both normal renal cells and cells of two renal carcinoma types expressed both ACE2 receptor and NRP1 co-factor required for SARS-CoV-2 entry. Then they verified these results by western blot analysis using normal and tumor renal cells. A correct (using a negative strand-specific RNA amplification) RT-qPCR analysis revealed that the same cell lines were permissive for virus, and in cases of high infection titers cytophathic effect is observed. The authors demonstrate formation of syncytia in renal tissue from a patient with previous COVID-19 infection, despite signs of the infection at a moment of analysis. In the discussion the authors speculate that SARS-CoV-2 can act as oncolutic viruses and may trigger tumor lysis in cancer patients with COVID-19, as observed for colon cancer patients.

The manuscript is carefully and clearly written. All conclusions are based on presented experimental data.

The topic definitely merits publication. Other groups have shown that SARS-CoV-2 does not always display cytophatic effect (i.e. in case of hepatocellular carcinoma Huh7 cell line).

I would suggest publishing the study after answering two short questions:

1. Was cytophatic effect quantified Ii.e. by MTT assay, staining with annexin V or propidium iodide)?

2. Can the authors speculate about the mechanisms that control cytopathic effect of the virus? or in authors opinion the effect is merely due to high replication rate? In our personal experience some cells lines survive during the infection, despite high replication rates.

6. PLOS authors have the option to publish the peer review history of their article (what does this mean?). If published, this will include your full peer review and any attached files.

Reviewer #1: No

Reviewer #2: No

---

## [Author Response · Author response to Decision Letter 0]

29 Sep 2022

The work and valuable constructive comments from the reviewers is highly appreciated and the text that follows is the same as in the response to reviewers file.

Response to the reviewer comments to the author

Reviewer #1: The study presented by Choong et al. implies that SARS-CoV-2 may replicate in renal cell carcinoma cells (CCRCC and PRCC) because these they express virus entry factors. Infection with SARS-Cov-2 resulted in cytopathic effect – syncytia formation. The manuscript is well written but I am afraid that some of the data presented suffer flaws and at some points there is overinterpretation of the results. 

Here are my comments concerning the study:

1. Line 71: the authors conclude that their data “provide insights into SARS-CoV-2 infectivity in renal cell carcinoma and define a potential viral entry mechanism that could be used to target renal carcinoma”. However, the latter is not shown experimentally nor sufficiently discussion in the Discussion section. Moreover, in the Discussion section, the authors widely stress on oncolytic viral therapy implicating SARS-CoV-2. Such notion is not supported experimentally. Do cancer cells strongly overexpress viral entry factors compared to normal cells? How will be the specific targeting of cancer cells achieved? Indeed, measles virus (the vaccine strain!) has been used in such therapies, but this is justified by the strong expression of MCP (CD46) in cancer cells and the non-pathogenicity of the virus which is not the case with SARS-CoV-2.

Response: We thank the reviewer for this comment. We agree that we have overstated the interpretation of our results and that a more measured approach serves the manuscript better. To this end we have rewritten line 70-71 (Our data provide insights into SARS-CoV-2 infectivity in renal cell carcinoma and define a potential viral entry mechanism that could be used to target renal cell carcinoma.). 

The new wording in line 70-71 is: Our data provide insights into SARS-CoV-2 infectivity in renal cell carcinoma and that the virus causes a distinct cytopathogenic effect.

We have also rewritten relevant parts of the discussion regarding oncolytic therapy in order to temper the conclusions. Specifically, we deleted the sentence beginning on line 346: (We suggest that the mechanism presented in this paper may be used in a similar fashion) and line 347-48: (Another possibility may be to use the receptor apparatus of a modified attenuated virus to deliver therapeutic nucleotides or a similar payload to the cancer cells). 

The new text is: “Whether SARS-CoV-2 uptake and propagation in RCC cells could elicit a similar response is uncertain. Usage of this uptake mechanism is questionable due to the the pathogenicitity of the virus and lack of selectivity due to expression in normal organs.” Line 356-359.

Based on bioinformatic analysis of the TCGA data set we do believe that RCC is unusual in its expression of the entry factors for SARS-CoV-2. This we can corroborate using tissue micro arrays of a large number of RCC cases. This in turn reflects the native expression of these factors in the proximal tubular epithelium of the kidney, a colocalization not shared by many organs. There are however other tissues expressing these factors, so it is not entirely specific and selective for delivery to RCC cells. The overly speculative concluding sentence of the discussion section has therefore also been deleted, line 350-351: (We suggest that the uptake mechanism for SARS-CoV-2 virus may be exploited to mediate uptake of therapeutic compounds or nucleic acids into renal cell carcinoma) and changed to: “The cytopathogenic effect causes RCC cell lysis and the delta-variant of the virus causes syncytialization of the cancer cells.”, now line 363-364.

2. Concerning the Discussion, the authors are not the first to show that SARS-CoV-2 entry factors are found in renal cells. This are just some examples of other studies: PMID:32903321 (TMPRSS2), PMID:19736548, PMID:11566082 (NRP1), and for ACE-2 there are multiple papers.

Response: We thank the reviewer for this point, we are aware of this, but regret that we have failed to communicate this in the manuscript. We do mention in the introduction, that the factors are expressed at mRNA level in kidney tissue, but regarding protein expression we were less clear. We have added the references PMID 11566082 and PMID 32903321 to the reference list in order to more clearly acknowledge previous histological work. The sentence describing this (line 90-92) has been changed to account for this. New sentence in line 92-93.

3. Figure 1C: I am afraid I do not see the expression of TMPRSS2 in normal tissues on the blot as stated by the authors: line 226. The authors also stated that “NRP1 expression was robust across all cases” (Line 228). This is not the case for CHRCC and some of PRCC samples. A better blot must be provided, and also a quantification of the blot by densitometry.

Response: We agree that the western blot for TMPRSS2 is definitely not the easiest to interpret. To amend for this, we tested alternative antibodies for TMPRSS2 on the protein lysates. We chose anti-TMPRSS2 from Thermo Fisher (1:1000, MA5-35756). This resulted in a much better blot with improved signal and we have replaced the inferior blot with the new one in a new Figure 1C. Information regarding this has been added to the materials and methods section in the manuscript (line 167-168). 

As suggested by the reviewer, we also performed densitometry of the bands in order to quantify ACE2, NRP1 and TMPRSS2, the results from this have also been added to figure 1C and to the figure legend (line 478-479). Finally, we have changed line 226-228 in the results section to reflect the improvements suggested by the reviewer. It now reads: 

The ACE2 protein was clearly detectable by three out of four CCRCC and PRCC cases and was undetectable in CHRCC lysates. NRP1 expression was strongly expressed by three of four CCRCC and PRCC cases, and at lower intensity by chromophobe RCC. Samples from CCRCC, PRCC and CHRCC expressed TMPRSS2 protein. (now line 229-232)

4. Figure 2A: Use arrows to indicate different histological structures.

Response: This suggestion improves evaluation of the images. For clarity figure 2A has been added the letter G to indicate glomeruli, arrows for examples of proximal tubules and asterisks for collecting ducts. This has also been added to the figure legend at line 485-486.

5. Line 246: “We also formalin” The verb is missing…

Response: we have changed the sentence to the more clear-cut sentence: “Cells from the CCRCC derived cell line RCC4 were also fixed by formalin and paraffin embedded.” This is now the text at line 251-253.

6. Figure 3B: The data of syncytia formation in PRCC is not convincing (the microscopic image). Moreover, there is no correlation between the cell culture data (slightly visible syncytia) and the PCR data (strong expression) for PRCC.

Response: We are grateful for the opportunity to clarify the figures. Supplemental Figure 3B does not show syncytia in cultured PRCC cells, it shows the formation of syncytia in cultured primary CCRCC cells. We agree that the image could be improved and we have therefore hatched the perimeter of two distinct syncytialised CCRCC cells in supplemental figure 3B and description added to figure legend (line 525). Judging from figure 1C and figure 3A, the levels of attachment proteins and SARS-Cov-2 replication seem roughly similar between normal tubular epithelial cells and primary CCRCC cells. The protein expression is slightly lower in the CCRCC cells, but not significantly so. 

7. I do not understand the including of the case study of 1 patient within this report. There is absolutely no proof that syncytia formation is due to SARS-CoV-2 (that the patient had 1 year before). Research for viral antigens was not performed. Moreover, syncytia formation is directly linked to cell lysis. This is even seen in the in vitro data on FigS3 B (almost no cells left).

Response: We agree with the concerns of the reviewer. We included the case, knowing that the number is too low to form a conclusion, especially since the active infection was a year previously. Having discovered that SARS-CoV-2 virus infects RCC cells in vitro, ideally, we would have liked to test the uptake in another system. The gold standard would be to infect a PdX mouse model with SARS-CoV-2. Since this is out of reach at this point, we reached out to the department of Urology asking if they had performed surgery on a patient with a proven SARS-CoV infection. They could only identifty one cases from their records. The histology being very unusual, we decided to add it to the paper in order to notify the pathology community about the possible morphology of a passing SARS-CoV-2 infection in RCC. The syncytia of the RCC-cases were definitely associated with cell lysis and outright necrosis of the RCC-tissue. This led us to design the experiment where we assessed if this could be repeated in vitro by a SARS-CoV-2.

As pointed out, we definitely understand the concerns of the reviewer and have added an additional sentence underscoring the speculative nature of the presented case at line 340-342 with the following wording: 

We are aware of the anecdotal nature of this RCC case and can therefore not draw any conclusions regarding potential effects in human RCC. 

We have also deleted the following sentence (line 331): We suggest that this finding strengthens our assumption that SARS-CoV-2 might infect RCC also in patients

8. In the discussion section (Lines 312-316) the authors say that “uptake and effect were not always clearly correlated to ACE-2 expression and that additional factors play role in causing cytopathogenic effect (CPE) in cell culture”. CPE is usually due to expression of viral receptors on uninfected cells and the expression of viral glycoproteins in infected cells. What is the expression of spike in these cells?

Response: This is a very interesting comment and suggestion. We would like to assess the expression of spike protein and accessory glycoproteins in the infected RCC and normal epithelial cells. However, we have not secured histological material or protein lysates from the infection assay, which precludes adding this data to the present study.

Reviewer #2: The manuscript by Choong et al describes a very promising phenomenon: oncolytic activity of SARS-CoV-2 against renal carcinoma variants. The authors used bioinformatic analysis of existing RNAseq data and showed that both normal renal cells and cells of two renal carcinoma types expressed both ACE2 receptor and NRP1 co-factor required for SARS-CoV-2 entry. Then they verified these results by western blot analysis using normal and tumor renal cells. A correct (using a negative strand-specific RNA amplification) RT-qPCR analysis revealed that the same cell lines were permissive for virus, and in cases of high infection titers cytophathic effect is observed. The authors demonstrate formation of syncytia in renal tissue from a patient with previous COVID-19 infection, despite signs of the infection at a moment of analysis. In the discussion the authors speculate that SARS-CoV-2 can act as oncolutic viruses and may trigger tumor lysis in cancer patients with COVID-19, as observed for colon cancer patients. The manuscript is carefully and clearly written. All conclusions are based on presented experimental data. The topic definitely merits publication. Other groups have shown that SARS-CoV-2 does not always display cytophatic effect (i.e., in case of hepatocellular carcinoma Huh7 cell line).

I would suggest publishing the study after answering two short questions:

1. Was cytophatic effect quantified Ii.e. by MTT assay, staining with annexin V or propidium iodide)?

Response: This is a very good point. We used the method preferred by the virologists at the virology department of the university hospital. Therefore, we assessed the cells by inspection and photography. The cytopathic effect was pronounced and easy to see and compared to infection controls and uninfected cells. Virus replication in cell culture as determined by visual cytopathic effect was further confirmed by RT-PCR. We did not run the suggested assays, partially due to infectivity issues.

2. Can the authors speculate about the mechanisms that control cytopathic effect of the virus? or in authors opinion the effect is merely due to high replication rate? In our personal experience some cells lines survive during the infection, despite high replication rates.

Response: For SARS-CoV-2 exposed kidney derived cell lines we observe a clear correlation between a high replication rate and CPE. For other cell lines such as calu3 more virus is necessary to induce CPE and a high viral replication rate.

A sentence regarding this has been added to the discussion section starting at line 330 with the following wording: 

The effect on the primary RCC cells was similar in magnitude to that of the kidney derived cell lines used to propagate SARS-CoV-2 virus in the clinical setting at the department of virology. Other cell lines used, such as calu3 require higher viral load for CPE and high replication (322-325).

In conclusion we hope that the changes and additions described above have met the expectations of the editor and the reviewers. We wish to thank all involved for their work in the constructive review of this manuscript.

---

## [Decision Letter · Decision Letter 1]

28 Oct 2022

PONE-D-22-06970R1SARS-CoV-2 replicates and displays oncolytic properties in clear cell and papillary renal cell carcinomaPLOS ONE

Dear Dr. Johansson,

Thank you for submitting your manuscript to PLOS ONE. After careful consideration, we feel that it has merit but would like to propose some further (very minor) changes, as outlined by reviewer 1 and in the editor comments further below, as we feel that these changes would further improve the manuscript.

We look forward to receiving your revised manuscript.

Kind regards,

Birke Bartosch

Academic Editor

PLOS ONE

Journal Requirements:

Additional Editor Comments:

The manuscript has been very positively reviewed. Some very minor issues remain which would further improve the manuscript. As suggested by reviewer 1, moving the CHRCC IHC from the supplementary into Figure 1 would make a good comparison of receptor expressing versus low/no expressing cells. Furthermore, if PCR data on CHRCC are available, it would be useful to include them into Fig 3A.

Reviewers' comments:

Reviewer's Responses to Questions

**Comments to the Author**

1. If the authors have adequately addressed your comments raised in a previous round of review and you feel that this manuscript is now acceptable for publication, you may indicate that here to bypass the “Comments to the Author” section, enter your conflict of interest statement in the “Confidential to Editor” section, and submit your "Accept" recommendation.

Reviewer #1: (No Response)

Reviewer #2: All comments have been addressed

2. Is the manuscript technically sound, and do the data support the conclusions?

Reviewer #1: Partly

Reviewer #2: Yes

3. Has the statistical analysis been performed appropriately and rigorously? 

Reviewer #1: N/A

Reviewer #2: Yes

4. Have the authors made all data underlying the findings in their manuscript fully available?

Reviewer #1: Yes

Reviewer #2: Yes

5. Is the manuscript presented in an intelligible fashion and written in standard English?

Reviewer #1: Yes

Reviewer #2: Yes

6. Review Comments to the Author

Reviewer #1: The authors have made changes in order to moderate some conclusions/statements which subsequently improves the manuscript. Even though, there are still issues that need to be addressed.

1. Figure 2: This figure includes IHC images of normal, CCRCC and PRCC cells but not CHRCC. It is more logical that S1A figure of CHRCC becomes part of Fig.2.

2. Figure 3A. All other cell types but CHRCC have been analyzed by PCR but CHRCC. If the authors have the data, it should be included (even if no infection is observed which could be expected).

3. I still have a problem with the case study of 1 patient. Lines 335-336: “We can only

suggest that SARS-CoV-2 may be causally connected to the unusual morphology”. This is a pure speculation. Was this patient treatment naïve prior nephrectomy? Nothing can be concluded from this “unusual morphology” of 1 patient. However, this oriented the discussion toward viral oncolytic therapy.

Reviewer #2: The authors have addressed all my concerns.

In my personal opinion, the findings that SARS-CoV-2 can replicate in clear cell carcinoma cells are important to understand permisiveness of cell to the pathogen, its tropism.

7. PLOS authors have the option to publish the peer review history of their article (what does this mean?). If published, this will include your full peer review and any attached files.

Reviewer #1: No

Reviewer #2: No

---

## [Author Response · Author response to Decision Letter 1]

7 Dec 2022

Thank you for the further reviews of our manuscript entitled: SARS-CoV-2 replicates and displays oncolytic properties in clear cell and papillary renal cell carcinoma (PONE-D-22-06970R1) by Oi Kuan Choong et al.

We are very grateful for the constructive comments and criticisms. Below we address the comments by reviewer #1. We interpret reviewer #2 as being satisfied with the current version. We believe that it has resulted in a further improvement of the manuscript. We hope that the reviewers find the measures taken adequate and that the current form of the manuscript is acceptable for publication.

Response to comments from Reviewer #1.

1. Figure 2: This figure includes IHC images of normal, CCRCC and PRCC cells but not CHRCC. It is more logical that S1A figure of CHRCC becomes part of Fig.2.

Response: We agree that it is more logical to have all the RCC stainings in one figure and have therefore moved the staining results for CHRCC from the supplemental figures to figure 2 D. We have also added H/E images of the same magnification as for figure 2A-C in order to achieve the same set up as for CCRCC and PRCC.

2. Figure 3A. All other cell types but CHRCC have been analyzed by PCR but CHRCC. If the authors have the data, it should be included (even if no infection is observed which could be expected).

Response: Regrettably, we could not show data from primary cultures of CHRCC. It would definitively have been appropriate as comparison. Primary CHRCC cells are difficult/

impossible to culture however, and we have not succeeded in establishing primary cultures of CHRCC cells. Also, there are no well characterized cell lines available to our knowledge. Therefore, we have no PCR data from the SARS-CoV-2 uptake and propagation experiment.

3. I still have a problem with the case study of 1 patient. Lines 335-336: “We can only suggest that SARS-CoV-2 may be causally connected to the unusual morphology”. This is a pure speculation. Was this patient treatment naïve prior nephrectomy? Nothing can be concluded from this “unusual morphology” of 1 patient. However, this oriented the discussion toward viral oncolytic therapy.

Response: One case is definitely not enough to infer that SARS-CoV-2 causes RCC cell necrosis. We sought to modify our wording in the previous version, and we now try to soften our language further. First, we have now omitted the sentence of above completely (“We can only suggest that SARS-CoV-2 may be causally connected to the unusual morphology” 

We also omit the sentence beginning at line 339, since we feel that is also a bit too bold: 

“The goal of precision medicine is to selectively target cancer cells. For well over a century, oncolytic viral therapy has been researched.”

We also add a new sentence to the discussion at line 346 to underscore that we are not implying a causal relationship. It reads as follows:

The isolated case described above does not prove causality for SARS-CoV-2 induced effects on CCRCC in patients. 

If the reviewer still expresses concerns, we are more than happy to completely omit the case from the article. The reason for not doing so before re-submission this time is because the finding of multinucleated cells in the cancer tissue led us to make an additional to find out whether the virus may cause syncytialization of the cells. Perhaps omission would render the assay hanging in the air so to say. Second, we wish to alert the readership to keep an eye on possible unusual histological patterns in this patient category. Again, we are more than happy to erase the case completely if its presence hinders the appreciation of the paper. Not least since the central theme of the article focuses on SARS-CoV-2 tropism in RCC and we do not want that this is obscured.

Summing up, we hope that our responses to the comments and suggestions from the reviewers are regarded as adequate and that they have resulted in an improved manuscript.

---

## [Editor Report · Decision Letter 2]

12 Dec 2022

SARS-CoV-2 replicates and displays oncolytic properties in clear cell and papillary renal cell carcinoma

PONE-D-22-06970R2

Dear Dr. Johansson,

We’re pleased to inform you that your manuscript has been judged scientifically suitable for publication and will be formally accepted for publication once it meets all outstanding technical requirements.

Kind regards,

Birke Bartosch

Academic Editor

PLOS ONE
---

## [Editor Report · Acceptance letter]

21 Dec 2022

PONE-D-22-06970R2 

SARS-CoV-2 replicates and displays oncolytic properties in clear cell and papillary renal cell carcinoma 

Dear Dr. Johansson:

I'm pleased to inform you that your manuscript has been deemed suitable for publication in PLOS ONE. Congratulations! Your manuscript is now with our production department. 

Kind regards, 

on behalf of

Dr. Birke Bartosch 

Academic Editor

PLOS ONE